# Spatiotemporal Dynamic Distribution, Regional Differences and Spatial Convergence Mechanisms of Carbon Emission Intensity: Evidence from the Urban Agglomerations in the Yellow River Basin

**DOI:** 10.3390/ijerph20043529

**Published:** 2023-02-16

**Authors:** Chaohui Zhang, Xin Dong, Ze Zhang

**Affiliations:** 1Faculty of Applied Economics, University of Chinese Academy of Social Sciences, Beijing 102488, China; 2Research Institute for Eco-Civilization, Chinese Academy of Social Sciences, Beijing 100710, China

**Keywords:** carbon emission, Yellow River Basin, urban agglomerations, spatial Markov chains, regional differences, spatial convergence model

## Abstract

Low-carbon transition is of great importance in promoting the high-quality and sustainable development of urban agglomerations in the Yellow River Basin (YRB). In this study, the spatial Markov chain and Dagum’s Gini coefficient are used to describe the distribution dynamics and regional differences in the carbon emission intensity (CEI) of urban agglomerations in the YRB from 2007 to 2017. Additionally, based on the spatial convergence model, this paper analyzed the impact of technological innovation, industrial structure optimization and upgrading, and the government’s attention to green development on the CEI’s convergence speed for different urban agglomerations. The research results show that: (1) The probability of adjacent type transfer, cross-stage transfer, and cross-space transfer of the CEI of urban agglomerations in the YRB is small, indicating that the overall spatiotemporal distribution type of CEI is relatively stable. (2) The CEI of urban agglomerations in the YRB has decreased significantly, but the spatial differences are still significant, with a trend of continuous increase, and regional differences mainly come from the differences between urban agglomerations. (3) Expanding innovation output, promoting the optimization and upgrading of industrial structure, and enhancing the government’s attention to green development has a significant positive effect on the convergence rate of the CEI of urban agglomerations in the YRB. This paper holds that implementing differentiated emission reduction measures and actively expanding regional collaborative mechanisms will play an important role in reducing the spatial differences in carbon emissions in urban agglomerations in the YRB, realizing the goals of peak carbon and carbon neutrality.

## 1. Introduction

With the continuous increase in carbon emissions becoming the focus of global attention, long-term carbon emission reduction targets have gradually been established worldwide. In the process of exploring the causes and mechanisms of high carbon emissions, cities and urban agglomerations have become the focus of research due to their high intensity of economic activities and energy consumption [1]. As far as China is concerned, the promotion of carbon emission reduction requires close attention to various densely populated urban agglomerations and major strategic regions with industries [2]. On the one hand, continuous urban construction and the converging of productive factors have brought significant carbon emission growth, and, on the other hand, the agglomeration effect has not generally produced a marked effect in promoting the improvement of carbon emission efficiency at this stage [3,4]. Combined with the development positioning and development status of different urban agglomerations, relevant studies have put forward different suggestions in terms of strengthening government regulation [5], optimizing spatial planning [6], adjusting the industrial layout [7], promoting technological innovation [8], etc., which provides scientific support for the promotion of carbon emission reduction and efficiency improvement in urban agglomerations. At present, the process of carbon control in large urban agglomerations such as Beijing–Tianjin–Hebei [9], the Yangtze River Delta [10,11], the Pearl River Delta [12], and other well-developed regions has achieved results. However, in other major strategic development regions and newly cultivated urban agglomerations located in central, central-western, and southwest China, the pressure for carbon reduction is still heavy.

Among major strategic development regions in China, the Yellow River Basin (YRB) is rich in fossil energy reserves and is an important energy and heavy chemical industry base in China. However, due to the centralized layout of traditional high-energy-consuming industries, the YRB is characterized by low development quality, difficult transformation, serious carbon emissions and pollution [13]. The ecology of the YRB is fragile, and the endogenous dynamic of development needs to be improved; promoting ecological governance and developmental transformation of the YRB has become a major issue that brooks no delay. In September 2020, China made a commitment to the world to strive to achieve peak carbon by 2030 and achieve carbon neutrality by 2060 at the 75th General Assembly of the United Nations. In March 2021, China proposed to deeply implement regional development strategy; the ecological protection and high-quality development of the YRB have become an important part of national long-term planning, and the strategic position of urban agglomerations in the YRB in the development of urban agglomerations across the country has been constantly improved. Under the background of development in the new era, the goals of peak carbon and carbon neutrality put forward new requirements for achieving ecological protection and high-quality development in the YRB. The necessity to solve the contradiction between economic growth and carbon emission growth in the development process of the YRB is increasingly prominent. Promoting low-carbon transformation has become an important driver of high-quality development in the YRB.

To provide a cognitive foundation for the implementation of scientific and reasonable carbon control policies in the YRB, some studies have conducted multi-scale and multi-angle analyses on issues related to carbon emissions in the YRB. From the overall perspective, the multi-region input–output model is used to calculate the sectoral carbon footprints of nine provinces in the YRB, and the transfer network of carbon emissions between different provinces and sectors is depicted [14]. From the scale-refinement perspective, some studies measure and analyze numerical change and spatial distribution pattern evolution in the energy consumption and carbon emissions of cities in the YRB based on night light data [15,16], making contributions to improving the accuracy of research on carbon emissions in the YRB. From the perspective of the multiple impacts of carbon emissions, some studies discussed the decoupling of urban carbon emissions in the YRB and its differentiated emission reduction paths and clarified the relationship between economic growth and carbon reduction pressure in different cities in the YRB [17,18] and the relationship between carbon emissions, ecological efficiency and economic growth [19]. Additionally, based on analysis of changes in the spatial agglomeration characteristics of carbon emissions in the YRB, the impact of various urban economic and social development factors on carbon emissions has been elaborated [20,21]. Furthermore, studies on carbon emission efficiency (CEE) and its influencing factors in the YRB are also increasingly enriching. On the other hand, current studies tend to focus on the calculation, spatiotemporal evolution, and impact factors of CEE [22], including the analysis of differences in CEE between resource-based and non-resource-based cities [23] as well as the spatial correlation and network structure of carbon emission efficiency among different urban agglomerations [24], enriching research content on carbon emission efficiency in the YRB. In addition, relevant research has also begun to focus on the analysis of carbon emissions in specific sectors in the YRB. Based on source differences in carbon emissions, some studies have measured the specific statuses of carbon emissions from land use [25], tourism [26], and transportation [27] in the YRB and analyzed the numerical changes in and influencing factors on different types of carbon emissions, which has promoted research on carbon emissions in the YRB to gradually expand into more subdivided fields.

Existing research basically covers analysis of the overall change trend and the main influencing factors on carbon emissions in the YRB. However, the development basis, development path and strategic positioning of each urban agglomeration in the YRB are different, and the results of carbon emission reduction in recent years are also different. To promote low-carbon transformation and high-quality development in the YRB, it is necessary not only to clarify the overall carbon emission conditions but also to conduct heterogeneity analysis from the perspective of urban agglomeration differences and coordinated development [28,29]. At present, there are few studies on distribution dynamics, differential changes and mechanisms of carbon emission reduction from the perspective of urban agglomerations in the YRB, and there is still a lack of empirical analysis on how to better promote the process of carbon emission reduction in urban agglomerations in the YRB. In view of the above, this paper uses the spatial Markov chain and Dagum’s Gini coefficient to characterize spatiotemporal dynamic evolution and changes in regional differences in carbon emissions in urban agglomerations in the YRB, and the spatial convergence model is used to analyze the convergence of carbon emissions for each urban agglomeration and the impact of carbon emission reduction conditions on the convergence rate in order to provide quantitative support for the coordinated development of urban agglomerations in the YRB and the achievement of peak carbon and carbon neutrality.

## 2. Materials and Methods

### 2.1. Study Area

The YRB is located in the new urbanization strategic pattern “Three Vertical and Two Horizontal” in China, and occupies a very important strategic position in China’s economic and social development and ecological security barrier construction. Urban agglomerations in the YRB are mainly composed of seven urban agglomerations [30]: Shandong Peninsula Urban Agglomeration (SDP), Central Plain Urban Agglomeration (CP), Guanzhong Plain Urban Agglomeration (GZP), Hohhot–Baotou–Erdos–Yulin Urban Agglomeration (HBEY), Urban Agglomeration along the Yellow River in Ningxia (NX), Jinzhong Urban Agglomeration (JZ), and Lanzhou–Xining Urban Agglomeration (LX).

At the municipal level, the developmental planning scopes of urban agglomerations at different levels are not identical; moreover, due to natural boundaries issues between cities and urban agglomerations in the YRB, there are always differences in the study areas of cities and urban agglomerations in the YRB [31]. In order to maintain the integrity of the administrative cell, 76 cities (prefectures) were finally included in the research scope of this paper, according to the “Planning Outline for Ecological Protection and High-quality Development of the YRB”, the “Development Planning of Central Plains Urban Agglomeration”, and other official urban agglomeration planning documents, on the basis of the geographical scope defined by the natural YRB and in consideration of the development radiation of each urban agglomeration (Figure 1).

### 2.2. Data Source and Processing

In terms of carbon emission data, this paper uses carbon emission data at China’s county level from 1997 to 2017 provided by the CEADs database. On this foundation, total urban carbon emission data for the YRB were obtained by merging data from the county level. The data on GDP were collected from the China City Statistical Yearbook, and the statistical caliber was at the municipal level. The missing data is supplemented by the Statistical Bulletin on National Economic and Social Development from various cities. Considering the availability and quality of data, this paper defines its research scope from 2007 to 2017. In addition, in order to eliminate the impact of price factors, this paper adjusts nominal GDP to the year of the initial sample (2007) through the GDP deflator index of the province to which each city belonged. Combing the urban carbon emission data and GDP, the urban carbon dioxide emissions per unit of GDP (CEI) was calculated.

### 2.3. Methods

#### 2.3.1. Spatial Markov Chain

In order to analyze the spatiotemporal evolution of the CEI of cities in the YRB, traditional and spatial Markov chain analysis was used to construct a Markov transition probability matrix. The traditional Markov chain is a Markov process with discrete times and states, emphasizing that the historical state is independent of the future state, which can be used to describe the evolution of the numerical objects over time, that is, the probability of evolution from state A to state B [32]. Due to the homogeneous mixing nature of carbon emissions, the evolution process of urban CEIs has the characteristic of “no aftereffect” geographically; that is, the CEI’s future stage is only related to the current stage, but not related to the historical stage, which exactly satisfies the requirement of the Markov chain method and also reflects good objectivity and rigorousness. In addition, different from the methods of time series and other index analysis methods that only focus on the change in the object value itself, the Markov chain method can also be used to infer the possibility of type transfer and has expandable geographical significance [33,34], which has unique advantages in analyzing the characteristics of CEI evolution. Therefore, this study first uses the Markov chain method to analyze the spatiotemporal evolution trend in overall CEI within the YRB.

Specifically, we discretized the continuous urban CEI data into k types according to a certain definition and then calculated the probability distribution and time variation of the corresponding types. Therefore, we could regard the dynamic evolution of urban CEI in the YRB as a Markov process. Generally, the urban carbon emission state at time t could be represented by a 1 × k state probability vector P_t_ = [p_1,t_, p_2,t_, …, p_k,t_]. The state transition probability for urban CEI in the whole time range could then be abstractly represented by k k × k Markov transition probability matrices, as follows:(1)$$p_{rh}=\frac{n_{rh}}{\sum_{t=1}^{k}n_{r,t}}\Rightarrow P=\left(\begin{array}{cccc} p_{11} & p_{12} & \cdots & p_{1h}\\ p_{21} & p_{22} & \cdots & p_{2h}\\ \cdots & \cdots & \cdots & \cdots \\ p_{r1} & p_{r2} & \cdots & p_{rh} \end{array}\right)$$

In Equation (1), *p_rh_* represents the probability that the city with state *r* at time *t* will transfer to state *h* at time *t + 1*; *n_rh_* represents the number of cities from state *r* at time *t* to state *h* at time *t + 1*; *n_r,t_* represents that the city is in state *r* at time *t*; *k* represents the states. In this paper, urban CEI is divided into four states according to quartiles, expressed by *k* = 1, 2, 3, 4 (low intensity, lower-middle intensity, upper-middle intensity, high intensity). Moreover, a transition of state from high intensity to low intensity is defined as “downward transfer”, and vice versa is defined as “upward transfer”. If there is no obvious transfer trend, it is defined as stable.

Furthermore, the spatial spillover effect caused by the process of interconnection and interaction between cities will have an impact on state transfer. Therefore, revealing the spatial spillover effect is of great significance for measuring the process of urban state transfer, and the spatial Markov chain can better describe the regional spatial spillover pattern [35]. The spatial Markov chain endows the traditional Markov state transition process with spatial delay conditions by incorporating the spatial weight matrix [36]. Specifically, the spatial Markov transition probability matrix decomposes the traditional Markov chain into *k*k* conditional transfer probability matrices based on the type of spatial lag for a certain area at time *t*, measuring the probability of a city’s transition state *r* at time *t* to state *h* at time *t* + 1 when the spatial lag is *k*.
(2)$$Lag_{i}=\sum L_{j}W_{ij}$$

In Equation (2), *i* and *j* represent cities in the YRB; *L_j_* represents the CEI value of city *j*; and *Lag_i_* is the spatial lag value of city *i*, which represents the weighted average of all spatially adjacent cities’ values. In this paper, the spatial weight matrix *W* was set as a geographic proximity matrix.

#### 2.3.2. Dagum’s Gini Coefficient and Decomposition

Regional differences, or the degree of regional inequality, are important objects in the analysis of economic and social indicators. Traditional and widely used inequality measurements indicators, such as the Thiel index and the classical Gini coefficient, are built on the assumptions of normal distribution and homoscedasticity, which strictly limit that there is no overlapping between the grouped samples and that it is difficult to decompose them into several sub-indexes with reasonable economic meaning [37]; that is, the difference contribution caused by an unreasonable grouping may be wrongly included in the contribution of a group or some groups, resulting in a deviation in judgment of the difference degree and difference contributions of these groups, while Dagum’s Gini coefficient can decompose overall differences in the sample into intra-regional differences, inter-regional differences, and inter-regional intensity in transvariation [38], which is beneficial to judge the sources and contributions of differences more clearly and is also a widely used difference measurement method with good analysis effects. The formula of inter-regional differences is as follows:(3)$$G_{ab}=\frac{\overset{i=1}{\underset{n_{a}}{\sum }}\overset{j=1}{\underset{n_{b}}{\sum }}\left|c_{ai}-c_{bj}\right|}{n_{a}n_{b}\left(C_{a}+C_{b}\right)}$$
where *a* and *b* refer to urban agglomeration groupings; *n_a_* and *n_b_* represent the number of cities in each urban agglomeration; *c_ai_* and *c_bj_* respectively represent the CEIs of city *i* in urban agglomeration *a* and city *j* in agglomeration *b*; and *C_a_* and *C_b_* represent the average CEIs of urban agglomerations *a* and *b*. If all cities in the sample are included in the same group, the Gini coefficient in this group is the Dagum’s Gini coefficient of the CEI of all cities. If the two urban agglomerations involved in the calculation are the same object (*a* = *b*), the result is the intra-group Gini coefficient *G_aa_* of urban agglomeration *a*.

Additionally, Dagum’s Gini coefficient can be decomposed into three parts as follows:(4)$$\begin{array}{l} G=\overset{k}{\underset{a=1}{\sum }}G_{aa}p_{a}s_{a}+\overset{k}{\underset{a=1}{\sum }}\overset{}{\underset{b\neq a}{\sum }}G_{ab}p_{a}s_{b}D_{ab}+\overset{k}{\underset{a=1}{\sum }}\underset{b\neq a}{\sum }G_{ab}p_{a}s_{b}\left(1-D_{ab}\right)\\ \Rightarrow GI+GB+GHB \end{array}$$
(5)$$\overset{}{\underset{a}{\sum }}\overset{}{\underset{b}{\sum }}p_{a}s_{b}=1;\begin{array}{cc} & p_{a}=n_{a}/N;\begin{array}{cc} & s_{a(b)}=n_{a(b)}C_{a(b)}/N \end{array} \end{array}C$$
(6)$$D_{ab}=\frac{d_{ab}-p_{ab}}{d_{ab}+p_{ab}}$$
(7)$$d_{ab}=E_{a}(cE_{b})+E_{b}(cE_{a})-E_{b}(c)$$
(8)$$p_{ab}=E_{a}(cE_{b})+E_{b}(cE_{a})-E_{a}(c)$$
where *p_a_* refers to the proportion of the number of cities in the urban agglomeration *a* (*n_a_*) relative to the total number of cities (*N*); *C* represents the average CEI of all cities; *s_a(b)_* refers to the ratio of the CEI of urban agglomeration *a* or *b* to that of all cities. In addition, the overall Gini coefficient is the weighted average of the Gini coefficient within the group (*G_ab_*) formed by the pairwise combination of all urban agglomerations, and the weight is *p_a_s_b_*. *D_ab_* represents the relative influence between urban agglomerations *a* and *b*, and the calculation process can be found in Equations (6)–(8). Before calculating *d_ab_* and *p_ab_*, the area codes of urban agglomerations *a* and *b* need to be adjusted to ensure *C_a_* > *C_b_*; *F_a_(·)* and *F_b_(·)* respectively represent the cumulative distribution functions of the adjusted CEI in *a* and *b*; *c* represents the overall CEI values, so *C_a_* and *C_b_* can be expressed as *E_a_(c)* and *E_b_(c)*, where *E* is the mathematical expectation calculation symbol. Therefore, *d_ab_* is the total influence between urban agglomerations *a* and *b*, which represents the desired value of all summary values between agglomerations *a* and *b* when *c_ai_* − *c_bj_* > 0; *p_ab_* represents the first-order moment of the intensity of transvariation between urban agglomerations *a* and *b*, which is the desired value of all summary values between agglomeration *a* and *b* when *c_bj_* − *c_ai_* > 0; *D_ab_* represents the proportion of differences in influence between two urban agglomerations (*d_ab_*-*p_ab_*) in the total maximum influence (*d_ab_* + *p_ab_*), and *D_ab_* is equal to *D_ba_*. When the scope of the two urban agglomerations completely coincide (*C_a_* = *C_b_*), *D_ab_* = *D_ba_* = 0. When the two urban agglomerations are completely different, *D_ab_* = *D_ba_* = 1. The Gini coefficient caused by the overlap between groups is called the inter-regional intensity of transvariation.

Therefore, *G_I_* represents the total contribution of the differences in urban CEI within each urban agglomeration to the overall difference, *G_B_* represents the contribution of the net differences between urban agglomerations, *G_HB_* represents the contribution of the inter-regional intensity of transvariation between urban agglomerations, and *G_B_* + *G_HB_* represents the total contribution of the differences between urban agglomerations. If the overall Gini coefficient *G* is higher, the overall differences in the CEI of urban agglomerations in the YRB will be larger. In addition, the smaller the contribution of the inter-regional intensity of transvariation between urban agglomerations, the smaller the contributions of regional differences caused by sample overlapping.

#### 2.3.3. Spatial Convergence Analysis

Convergence analysis is a classical method used in the field of economic growth theory to analyze the gaps between countries or regional economic development indicators and their dynamic trends [39]. From economic convergence to environmental convergence and then to carbon emission convergence, convergence theory has gradually established a connection with carbon emissions through the analysis of the relationship between income and pollution in the Kuznets curve [40]. So far, the convergence model has been widely used to analyze the convergence status of carbon emissions in a country (or region or city) and combined with various econometric models to analyze the factors affecting the convergence rate as well as to judge the future numerical change trends in the target object to a certain extent [41,42]. Additionally, compared with other econometric models, the convergence model has more intuitive numerical significance and purposiveness. Therefore, as to the YRB’s carbon emissions issues, some studies used the convergence model to analyze CEE gaps within the YRB, convergence differences in CEI between the YRB and other large urban agglomerations, and the main influencing factors that promote the improvement of CEE or lead to a decline in CEI [37,43], which provides a basic explanatory reference for subsequent research.

In view of above, this paper uses σ convergence and β convergence to analyze whether the CEI of urban agglomerations in the YRB converges to a steady state over time, and examine the convergence status and speed combined with the spatial econometric model.

σ convergence refers to the trend that the dispersion of CEI of each urban agglomeration decreases with time. If the σ value decreases, the dispersion of CEI in the urban agglomeration is decreasing over time; that is, the internal level differences in CEI are narrowing with the σ convergence phenomenon [44]. In this paper, the coefficient of variation is used to measure the convergence value, as follows:(9)$$\sigma =\frac{\sqrt{\sum_{i=1}^{n_{a}}(c_{ai}-C_{a}{)}^{2}/n_{a}}}{C_{a}}$$
where *c_ai_* represents the CEI of city *i* in urban agglomeration *a*; *C_a_* is the average CEI of urban agglomeration *a*; *n_a_* is the number of cities in urban agglomeration *a*.

β convergence refers to the fact that, with time, regions with higher CEI will have a greater decline to catch up with regions with lower CEI, and the gap between the two will gradually narrow, eventually reaching the same stable level. β convergence can be divided into absolute β convergence and conditional β convergence [45].

In this paper, absolute convergence refers to the convergence trend of the CEI of the target region without considering a series of factors that have an important influence on the CEI, as follows:(10)$$\ln\left(\frac{c_{i,t+1}}{c_{it}}\right)=\alpha +\beta \ln\left(c_{it}\right)+\mu_{it}$$
where *c_it_* and *c_i,t+1_* represent the CEI value of city *i* at time *t* and *t + 1* respectively; *ln*(*c_i,t+1_*/*c_it_*) represents the growth rate of CEI in city *I* at time period *t + 1*; and β is the convergence coefficient. If β < 0, there is a convergence trend in regional CEI and vice versa. The convergence speed (*v*) can be calculated by *v* = *−ln(1 − |β|)/T*; *μ_it_* represents random disturbance.

Considering the increasing interaction between cities and the cross-regional flow of resource elements, it is necessary to bring spatial effects into analyzing the convergence in the CEI of urban agglomerations in the YRB. Therefore, the spatial econometric model of absolute β convergence can be constructed [46] as follows:(11)$$\begin{array}{l} \ln\left(\frac{c_{i,t+1}}{c_{it}}\right)=\alpha +\beta \ln\left(c_{it}\right)+\rho \overset{j=1}{\underset{n}{\sum }}w_{i,j}\ln\left(\frac{c_{i,t+1}}{c_{it}}\right)\\ \;+\gamma \overset{j=1}{\underset{n}{\sum }}w_{ij}\ln\left(c_{i,t}\right)+\mu_{it},\mu_{it}=\lambda \sum_{j}w_{ij}\mu_{it}+\varepsilon_{it} \end{array}$$
where *t* represents year; *i* and *j* refer to city units; and *w_ij_* refers to the element in the spatial weight matrix. In this paper, the geographic distance matrix is set as the benchmark spatial weight matrix; ρ and γ are both spatial lag coefficients; λ represents the spatial error coefficient; and μ is random disturbance. If λ = 0, Equation (11) is a spatial Dubin model. If λ and γ are both equal to 0, Equation (11) is a spatial autoregressive model (SAR). If ρ and γ are both equal to 0, Equation (11) is a spatial error model (SEM). Moreover, although the spatial Dubin model is more general and has fewer missing factors than SAR and SEM, the applicability of SDM, SAR, and SEM depend on strict statistical tests.

Furthermore, on the basis of the absolute β model, conditional β convergence includes a series of important affecting factors to examine whether the CEI will still converge to its steady state and the change of its convergence rate.
(12)$$\begin{array}{r} \ln\left(\frac{c_{i,t+1}}{c_{it}}\right)=\alpha +\beta \ln\left(c_{it}\right)+\rho \overset{j=1}{\underset{n}{\sum }}w_{ij}\ln\left(\frac{c_{i,t+1}}{c_{it}}\right)\\ +\gamma \overset{j=1}{\underset{n}{\sum }}w_{ij}\ln\left(c_{it}\right)+\delta X_{i,t+1}+\mu_{it} \end{array}$$
where X represents the control variable vector and δ represents the parameter vector to be estimated. The change conditions of the different forms of spatial econometric model in Equation (12) are the same as in Equation (11).

Considering the series of measures taken under the peak carbon dioxide emissions and carbon neutrality goals on the one hand, it is of great necessity to use technological progress as the core driving force, and use energy transformation to force industrial transformation to improve development efficiency and quality. On the other hand, the low-carbon transition also reflects long-term considerations of climate change, maintaining green ecology, and promoting sustainable development. Against the overall background of development transformation, coordinated promotion of emission reductions, and environmental improvement, the logic of China’s economic growth has undergone profound changes, and the coordinated advancement of “carbon reduction, pollution reduction, green expansion, and economic growth” is imperative.

Although past relevant research shows that energy structure, trade status, openness degree, and some demographic and urban construction factors are also important factors affecting carbon emissions, as far as this article is concerned, focusing on major emission reduction factors and conducting a comprehensive and rigorous econometrics analysis may lead to more targeted conclusions; particularly, it is necessary to observe the effect of the inclusion of different factors on the convergence rate. Moreover, in order to better conduct an in-depth analysis on the basis of data analysis in the first half of the article, as well as adopting more targeted and innovative influencing factors to obtain more profound research conclusions, this study takes “main endogenous power”, “industrial adjustment and optimization status” and “government’s tendency” as the points of penetration, and includes the following three specific carbon reduction mechanisms as conditions to analyze their impact on the spatial convergence and convergence speed of the CEI of the urban agglomerations in the YRB:

(1) Urban innovation. Innovation-driven development is the core driving force in the high-quality development of urban agglomerations, and it is also the key to promoting low-carbon transformation. This paper uses the ‘urban innovation index’ as the measurement index for urban innovation (INNOV). The data comes from the report “China’s Urban and Industrial Innovation 2017” prepared by the Industrial Development Research Center of Fudan University [47], mainly based on updated information on the legal status of micro-patents for inventions and the annual fee data for different patents for inventions authorized by the State Intellectual Property Office. This indicator is obtained by summing up and standardizing city-level data by industry, and it has the advantage of measuring and characterizing innovation input and innovation output. The greater the value of the urban innovation index, the stronger the urban innovation ability.

(2) Optimization and upgrading of industrial structure. The focus of low-carbon transformation is to promote low-carbon development and overall economic and social progress by relying on technological upgrading and industrial upgrading. Therefore, promoting the upgrading and optimization of industrial structures is the key path to carbon emission reduction. If the overall industry is relatively low-carbon and high-value-added, and the input of a unit of energy can produce a higher economic output, the intensity of carbon emissions will certainly decline, and the total carbon emission will also tend to stabilize. Existing research mainly measures the optimization and upgrading of industrial structure through the advancement of the industrial structure index and the industrial structure rationalization index. The former is often used to measure industrial structure upgrading, to reflect the trend and process of the overall quality and efficiency of the industrial structure evolving from a lower level to a higher level, while the latter is used to measure the degree of optimization and coordination among industries and the degree of effective utilization of resources. This paper considers including these two indicators at the same time to comprehensively characterize the industrial optimization and upgrading of the urban agglomeration in the YRB.

The industrial structure upgrading index (ISU) is represented by calculating the layer coefficient of the industrial structure [48], as follows:(13)$$ISU=\sum_{i=1}^{3}m*I_{i}$$
where *ISU* represents the upgrading of the industrial structure, *I_i_* (i = 1, 2, 3) represents the proportion of output value of the primary, secondary and tertiary industries of GDP; m (m = 1, 2, 3) represents the weight allocated to these industries.

The reciprocal of the Theil index is used to measure the industrial structure rationalization index (ISR) in order to take into account the structural deviation of output value and employment in different industries, as well as the different economic status of each industry at the same time [49,50], as follows:(14)$$ISR=\frac{1}{TI}=\frac{1}{\sum_{i=1}^{n}\frac{Y_{i}}{Y}\ln\left(\frac{Y_{i}}{L_{i}}/\frac{Y}{L}\right)}$$
where *TI* represents the Theil index; *Y_i_* (i = 1, 2, 3) represents the total value of the primary, secondary, and tertiary industries; *Y* represents the total value of all industries; *L_i_* represents the number of people employed in each industry; *L* represents the total number of employees of all industries. The greater the *TI*, the lower the degree of industrial structure optimization, and vice versa. Therefore, a higher ISR indicates a more optimized industrial structure.

(3) Government’s attention to green development. Since September 2020, when China put forward the goals of peak carbon and carbon neutrality, the process of economic and social development transforming into green and low-carbon development has been fully started, and carbon emission reduction has also become the focus of China’s ecological and environmental protection. Considering that China’s energy structure is still dominated by fossil energy, with high carbon emissions, and that there is a high homology among carbon emissions and other pollutant emissions, plus that carbon emissions and carbon emission reduction behavior involve strong externalities, the government’s guiding role in green and low-carbon development is crucial. This study collected the government work reports for cities in urban agglomerations in the YRB during the research period and counted the number of words related to “ecology, green, low-carbon, and environmental protection” by using the text analysis method through Python. Furthermore, it calculated the proportion of “relative word frequency” to measure the government’s attention to green development (*AGD*), as follows:(15)$${AGD}_{it}=\frac{f_{it}/f_{it,total}}{F_{it}/F_{it,total}}$$
where *AGD_it_* represents the degree of attention to green development of city *i* at time *t*, *f_it_* represents the number of words related to green development in the annual government work report of city *i* at time *t*, *f_it,total_* represents the total number of words in the annual government work report of city *i* at time *t*, *F_it_* represents the number of words related to green development in the annual government work report of the urban agglomeration to which city *i* belongs, and *F_it,total_* represents the total number of words in the annual government work report of the urban agglomeration where city *i* belongs.

In addition, considering the possible impacts of urban economic development level, population convergence, and infrastructure construction on carbon emissions, this paper also includes per-capita GDP, population density, and road area ratio as control variables. The data come from the China City Statistical Yearbook and the Statistical Bulletin of each city. All variables related to price are adjusted to the actual value of the base period (2007). Some missing data are interpolated using the trend forecasting method. Variable interpretation and descriptive statistics are described in Table 1.

## 3. Results

### 3.1. Spatiotemporal Distribution Dynamics of CEI of Urban Agglomerations in the YRB

#### 3.1.1. Spatiotemporal Distribution Characteristics

Combined with the changes in the distribution position, the distribution form of the main peak, the degree of tailing, and the number of wave crests of the kernel density curve during the study period, it can be found that there is a significant left shift trend of the distribution position of the nuclear density curve, indicating that there is a downward trend in the CEI of the urban agglomerations in the YRB (Figure 2a). The height of the main peak decreases, and the width becomes larger, indicating that the differences in CEI within urban agglomerations continue to rise. The significant right-tailing and double-peak phenomenon in the period indicate that the CEIs of some cities in the urban agglomerations are significantly higher than that of other cities. Secondly, from the changes in the average CEI of each urban agglomeration, the average CEI of NX is the highest, followed by JZ, and both of them exceed the overall average level. The average CEI of LX is basically equal to the average level, while the average levels of SDP and CP are relatively low. It is worth noting that, when the overall CEI is decreasing, the rate of reduction is also slowing down. The average CEI of all cities and the urban agglomerations of NX and LX rebounded at the end of the study period. Finally, from the spatial distribution and standard deviation ellipses of the CEIs of each urban agglomeration (Figure 3), although there is a downward trend in the CEIs of urban agglomerations in the YRB generally, the areas and rotation angles of the standard deviation ellipses have no significant changes, indicating that the spatial distribution mode of CEI of urban agglomerations in the YRB has not changed significantly from 2007 to 2017. The center of gravity of the ellipse moves slightly from the middle to the west, indicating that the rate of decline of urban CEI in the western cities in the YRB is slightly lower than the overall average level.

The spatiotemporal distribution characteristics of the CEI of urban agglomerations in the YRB are essentially determined by the development characteristics and development level of the urban agglomerations themselves. Among the seven major urban agglomerations in the YRB, developmental maturity shows an obvious increasing trend from the upstream to downstream. This can also be inferred from the developmental orientations of different urban agglomerations. For instance, NX is a national heavy energy and chemical industry and new materials base, JZ is a national energy base, advanced manufacturing base, and resource-based economic transformation demonstration zone, and SDP is an important opening gateway in the north; the national “Blue Economic Demonstration Zone” and the “High-efficiency Ecological Economic Zone” hold totally different development statuses comparatively. In a common view, due to resource endowment, historical industrial layout, and other geographical factors, the closer to the upper reaches of the Yellow River, the lower the population density, and the lower the overall development level of the urban agglomeration. Relatively small economic aggregates and lower innovation capacity lead to a lower ability for upstream urban agglomerations to address the pollution and emissions caused by traditional industries. The low development efficiency, low ability to control carbon emissions, and increasingly serious carbon emission problems form a negative cycle. Besides, although the proportion of traditional manufacturing industry in the economic structure of the urban agglomerations in the YRB is generally large, and the proportion of advanced manufacturing industry and modern service industry is generally low, the developmental differences in different urban agglomerations determine the difficulty of achieving the carbon emission reduction target. Urban agglomerations with better development status have gradually controlled CEI to a relatively low level, while urban agglomerations with high CEI remain in the high-CEI range even if their absolute emission reduction is relatively high, that is, if the difference in CEI does not decrease. Therefore, even under the general trend of carbon reduction, the spatiotemporal distribution pattern, especially the center-of-gravity-shift character of the CEI values of urban agglomerations, has not changed significantly.

#### 3.1.2. Spatiotemporal Transfer Characteristics

According to the spatial distribution of the traditional Markov transition probability matrix and its transfer (Figure 4), the overall type transfer tendency of urban CEI in the YRB is obvious. Firstly, the probability on the diagonal is far greater than that of the other area in the matrix, and the minimum probability of maintaining the original state is 85.71% throughout the study period, which indicates that the type of urban CEI in the YRB has strong stability on the whole, and the possibility of state transfer is small. Secondly, the stabilities (85.71%, 87.13%) of lower-middle intensity and higher-middle intensity on the diagonal are significantly lower than those of low intensity and high intensity at both ends (94.05%, 91.76%), indicating that there is a higher probability for cities with bipolar-style CEI to maintain their respective CEI types all the time. From the probabilities on both sides of the diagonal, the probability of upward transfer for low-intensity cities is only 5.95%, meaning that the possibility of low-carbon emission cities deteriorating is low. However, the probability of downward transfer for high-intensity cities in the initial year is 8.24%, which indicates that this type of city is in a “path lock” state. In addition, the probability of downward transfer for lower-middle intensity cities (8.57%) is higher than that of upward transfer (5.71%), while the reverse applies to upper-middle cities, indicating that there is a greater probability of upper-middle cities maintaining higher CEI; Thirdly, in the transition probability matrix, the elements with a probability of 0 are far away from both sides of the diagonal, which indicates that the probability of adjacent type transfer of urban CEI in the YRB is small and the possibility of cross-stage transfer is also small.

In order to further identify the regular pattern of dynamic transfer of urban CEI in the YRB in time and space, this paper analyzes the impact of different neighborhood types on the transfer of urban CEI type by constructing a spatial Markov transfer probability matrix. First, it is necessary to test the spatial correlation of urban CEI in the YRB. The global Moran’s I index of urban CEI in the YRB from 2007 to 2017 is significantly positive at the level of 1%, indicating a significant positive spatial correlation of the explained variables (Table 2). In addition, there is an increasing trend in Moran’s I index with some fluctuations, indicating that the spatial concentration of urban CEI in the study period shows a certain trend of improvement. However, the overall fluctuations are small, indicating that the spatial concentration level of urban CEI in the YRB has a small change.

Besides, it is necessary to test the spatial spillover effect of CEI transfer in the YRB. To test whether the CEI’s spatial spillover effect for cities with different types of neighborhood backgrounds is statistically significant, it can be assumed that ‘the type transfer of CEI of cities in the YRB during the study period is independent of each other, independent of the type of spatial lag’, and then the chi-square statistic can be constructed to test the Markov transition probability with spatial lag conditions [51], as follows:(16)$$S=-2\log\left\{\prod_{L=1}^{k}\prod_{x=1}^{k}\prod_{y=1}^{k}{\left[\frac{p_{xy}}{p_{xy}(Lx)}\right]}^{nxy(L)}\right\}$$
where *k* represents the type of CEI; *p_xy_* represents the traditional Markov transition probability; *p_xy_*(*Lx*) and *n_xy_*(*L*) represent the spatial Markov transition probability and the corresponding number of cities when the type of spatial lag is *L* (*L* = 1, 2, 3, 4; namely low, lower-middle, upper-middle, and high CEI); and *S* is subordinated to the chi-square distribution when the degree of freedom (df) is *k*(*k* − 1)^2^. After calculation, when the degree of freedom is not adjusted, the chi-square statistic is 74.554 (*p* = 0.000), which rejects the original hypothesis, indicating that there is a significant spatial correlation between the type of CEI and the type of neighborhood, and it can be considered that the different neighborhood types have a significant impact on the type transition of CEI of cities in the YRB.

By taking the type of neighborhood as the condition and adding it to the traditional Markov transition probability matrix, the spatial Markov transition probability matrix can be obtained. The type of neighborhood does have a significant impact on the state transfer of urban CEI in the YRB, and different neighborhood types have different impact directions on the regional state transfer. From a general view, if a city is adjacent to an area with low CEI, the probability of its CEI transferring upward will increase, and the probability of its CEI transferring downward will decrease; that is, the neighborhood state of low CEI will play a positive role in the city’s emission reduction, and vice versa.

From the perspective of specific probability changes (Figure 5), first of all, if the neighborhood is not considered, the probability of a high-CEI city’s moving downward is 8.24%. If the city is in a high-CEI-neighborhood state, the probability of downward transition will decrease to 3.33%, but if it is in a low-CEI-neighborhood state, the probability of downward transfer will increase to 11.86%; The probability of downward transfer of a ‘low-intensity’ type city is 5.95%. If it is in a ‘high-intensity’ neighborhood state, the probability of downward transfer will decrease to 1.82%, indicating that cities with low CEI are more sensitive to their neighbors with high CEI and are more exposed to negative impacts. If a city with low CEI is in a high-CEI-neighborhood state, its downward transition probability will drop to 1.82%, indicating that cities with low CEI are more sensitive to their neighbors with high CEI and are more exposed to negative impacts. Secondly, for cities with low CEI, the probability of downward transition (8.57%) is higher than that of upward transition (5.71%), and the probability of downward transition increases to 13.95% when one city with low CEI is adjacent to another city with a similar state, while the probability decreases to 7.69% when a city with low CEI is adjacent to a city with high carbon emissions. For cities with high CEI, the probability of downward transition (4.68%) is less than that of upward transition (8.19%). When the CEIs of two cities are high, and the cities are adjacent to each other, the probability of downward transition will decline to 2.27%, and the probability of maintaining the original state will increase significantly. In general, the urban CEI of the YRB has significant spatial agglomeration and spatial spillover effects. That is, when a city is in a region with high CEI, the probability of CEI rising will increase, while, if it is in a region with low CEI, the probability of CEI decreasing will increase. However, it should also be noted that the state of urban CEI in the YRB remains relatively stable after incorporating the neighborhood conditions. The tendency of changes in urban CEI tends to be consistent, and the elements are also far away from both sides of the diagonal when the probability of spatial state transition matrix is equal to 0, indicating that the possibility of cross-stage change and cross-space change in urban CEI in the YRB is small.

From the overall data performance during the study period, the CEI of cities in the YRB has an obvious “path lock” character; that is, the CEI type will not change significantly with time. Although the transfer probability changes in line with expectations after the spatial neighborhood conditions are included, the overall change is still small, which also confirms the conclusions of the previous analysis. Without prejudice, if the endogenous power of carbon emission reduction and control remains unchanged, it is difficult for a city with high CEI in the YRB to achieve the carbon emission reduction goals, even if it has a few “model neighbors” with low carbon emissions and higher development levels. Furthermore, similar to the reason for there being no obvious changes in spatiotemporal distribution, the spatiotemporal transfer of CEI maintains strong stability, mainly because the CEI difference between different cities has not decreased significantly, as the result of the Markov chain method shows—the number of cities at different quantiles of CEI has not changed significantly.

### 3.2. Regional Differences in CEI of Urban Agglomerations in the YRB

#### 3.2.1. Differences within Urban Agglomeration

The spatial Markov transition probability matrix reveals the overall spatiotemporal evolution trend in urban CEI in the YRB, and the relevant conclusions indicate that it is of necessity to analyze differences in CEI in YRB urban agglomerations specifically. Therefore, this study further identifies the differences and the changes in different urban agglomerations in the YRB by calculating Dagum’s Gini coefficient.

From the perspective of all cities and the internal differences in the urban agglomerations (Figure 6), first of all, the overall Gini coefficient of the urban CEI in the YRB is relatively high, with an average value of 0.27 in the study period, maintaining a growth trend, reaching 0.31 by 2017. This shows that there is an obvious imbalance in urban CEI in the YRB, and the imbalance degree is deepening. From the perspective of each urban agglomeration, the Gini coefficient within all urban agglomerations does not exceed the overall Gini coefficient, which indicates that the differences within each urban agglomeration are not the main source of the overall differences, but the variation trend in the internal differences within each urban agglomeration still shows a large difference. Secondly, the internal differences in the CEI of GZP are the largest, with an average of 0.24 during the period, showing a strong upward trend after 2013. The reason is that there are many cities whose CEI is generally lower than other cities in GZP, such as Xi’an and Xianyang, and this high internal difference also indicates that the polarization of the CEI of GZP is intensifying. The internal difference in the CEI of JZ is lower than that of GZP, with an average value of 0.22 in the period, but the overall change is small, indicating that the CEI of JZ has always maintained a stable internal difference level and the polarization phenomenon is not obvious. Similar to that of the GZP, the main source of the internal difference in the CEI of JZ is that the carbon emissions of provincial capitals are generally lower than that of other cities. Thirdly, the internal difference in the CEI of LX presents an obvious fluctuation and trend. It shows a large upward trend during the period, but the difference level at the beginning of the period is basically the same as that at the end of the period, indicating that the status quo of high carbon emissions and high difference has been controlled and alleviated in recent years. Moreover, the internal difference for CP basically remains at the average level of 0.15, and the changes of difference are small, indicating that the internal difference in CEI is stable and non-polarized. The internal difference for HBY is the smallest, with an average of 0.09 during the period, but there is an obvious upward trend in difference during the period, indicating that the progress gap in carbon emission reduction within the agglomeration has consistently enlarged. Furthermore, as the urban agglomeration with the highest CEI in the YRB, the internal differences in NX are relatively low. Although the absolute value of each city’s CEI in NX is decreasing, which shows that the pace of carbon emission reduction among cities is relatively consistent, cities that can bear the main responsibility of carbon reduction are still not cultivated within the urban agglomeration, and the overall pressure for emission reduction is still large. Compared with NX, SDP, the urban agglomeration with the lowest CEI in the YRB, has low internal differences and a downward trend, indicating that its emission reduction effect is stable.

#### 3.2.2. Differences between Urban Agglomerations

As to the differences between urban agglomerations (Figure 7), first of all, the differences in carbon emissions between urban agglomerations in the YRB mainly show obvious differences in the east, west and east regions, and the overall differences between urban agglomerations show an increasing trend. Specifically, the regional difference between NX and other urban agglomerations is the largest and still maintains an increasing trend. The average difference between NX and SDP is 0.58, and the difference level reached a peak of 0.658 in 2017. In addition, the regional difference between JZ and other urban agglomerations is relatively high, followed by the regional difference between GZP and other urban agglomerations, indicating that the gap between the two midstream urban agglomerations and other urban agglomerations in terms of carbon reduction is also obvious. The difference between HBEY and other urban agglomerations is relatively small, and the average value in the period does not exceed 0.25. The difference between CP and HBEY is the smallest, with a mean of 0.15 in the period, which shows that the spatial difference in the CEI of these urban agglomerations is relatively small.

#### 3.2.3. Contribution of Differences

Dagum’s Gini coefficient can decompose the overall differences in the CEI of urban agglomerations in the YRB into three parts, including the contributions of intra-regional differences, inter-regional differences, and inter-regional intensity of transvariation. From the absolute value of each part’s contribution and proportion (Figure 8), there is a downward trend in the intra-regional Gini coefficient of the CEI of urban agglomerations in the YRB, with some fluctuations. The average change rate of the contribution value in the period is only 0.67%. The contribution to the overall difference decreases from 19.91% at the beginning of the sample to 13.83% at the end of the sample. The average contribution rate of the total difference between urban agglomerations is 82.78%, which indicates that inter-regional difference is the main source of the differences in CEI of urban agglomerations in the YRB. This is consistent with the previous conclusions. The contribution of the net difference between urban agglomerations is 0.16 at the initial stage of the sample and 0.23 at the end of the sample, respectively, with an average increase of 4.13%. Its contribution to the overall difference also rose from 67.17% to 75.87%, and the average contribution rate during the sample period reached 71.15%. Additionally, the contribution value and contribution rate of the inter-regional intensity of transvariation among urban agglomerations are the lowest, with the contribution value ranging from 0.28 to 0.33, and the average contribution rate during the period is 11.62%. The inter-regional intensity of transvariation reflects the contribution of the overlapping parts of each sub-sample group to the overall differences, and the small proportion indicates that the regional division of urban agglomerations can effectively distinguish cities with different carbon emission states, reducing the impact of inter-group overlapping on the difference decomposition results.

### 3.3. Spatial Convergence Analysis of the CEI of Urban Agglomerations in the YRB

#### 3.3.1. σ Convergence

The evolution trend of σ convergence of the CEI of urban agglomerations in the YRB from 2007 to 2017 is as follows (Figure 9). From all cities, the coefficient of variation of CEI shows an obvious upward trend. The lowest point occurs at the beginning of the period, and the ending value is higher than the initial value, which means σ convergence characteristics do not exist, which also confirms the conclusion that the overall gap in carbon emission reduction in urban agglomerations in the YRB is expanding. In terms of urban agglomerations, there is a slight σ convergence trend for SDP. The ending value of the variation coefficient of CEI for other urban agglomerations is greater than the initial value; that is, σ convergence does not exist. This shows that, except for in the SDP, there is not a significant downward trend in spatial difference in the CEI of urban agglomerations in the YRB.

#### 3.3.2. Absolute β Convergence

Since the spatial effect of CEI may be different at different spatial scales, this paper first uses the LM test to determine whether there is spatial autocorrelation of the absolute β convergence of the CEI of urban agglomerations in the YRB (Table 3). Wald and LR tests are then used to select the optimal spatial model. As to the absolute convergence characteristics of urban agglomerations in the YRB in Table 4, first of all, there is absolute β convergence for cities and urban agglomerations in the YRB, indicating that there is a trend of decline and convergence to their respective stable levels in the CEIs of urban agglomerations in the YRB in the long run, without considering the impact of convergence conditions on CEI. Secondly, the spatial lag coefficients of the growth rates of CEI for urban agglomerations in the YRB are significantly positive at the level of 1%, which indicates that the growth rates of CEI for cities will be affected by the positive spatial spillover of neighboring cities at each urban agglomeration scale and under each model. However, spatial effects also show different patterns. Among them, there is spatial lag for both the growth rate term and convergence term of CEI for all cities, SDP, and CP. The spatial lag of the error term exists in GZP and JZ. There is only a spatial lag of the growth rate term in NX, HBEY, and LX. Furthermore, the convergence rate for urban agglomerations in the YRB is different. Among them, the convergence rate for NX is 0.0241, which is higher than the average level of the urban agglomerations in the YRB, as the average CEI of NX is the highest in the period and the absolute value of its carbon emission reduction is the largest. The convergence rates of SDP, CP, and JZ are fast and similar in value (the average convergence rate is 0.0171), indicating that the reduction rates of carbon emissions for these three urban agglomerations are similar. The convergence rates for GZP, HBEY, and LX exhibit slow with similar values (averaging at 0.0088), which indicates that the reduction rate for carbon emissions is comparatively slow and the reduction amplitudes are similar.

#### 3.3.3. Conditional β Convergence and Carbon Emission Reduction Effect

The results of absolute β convergence reveal the convergence rates of the CEI of cities in the YRB, while the changes in convergence rate can be further investigated under different carbon emission reduction conditions through conditional β convergence. The LM test shows the existence of a spatial autocorrelation effect of each model (Table 5). The results of Wald and LR tests show that SDM will not degenerate under each variable. In terms of all cities (Table 6), first of all, the convergence coefficient β of CEI under various conditions is significant at the level of 1%, indicating that conditional β convergence exists. Furthermore, compared with the speed of absolute β convergence, the speed of conditional β convergence is obviously increasing, indicating that the included mechanisms have a significant positive impact on carbon emission reduction in the YRB. In terms of specific conditions, the industrial structure rationalization index plays the most significant role in improving the convergence rate of CEI, and the rate of conditional β convergence has increased by 60.94% compared with the rate of absolute β convergence, which means that improving the transformation ability of the industrial structure and the effective utilization of resources will certainly play a significant role in promoting CEI convergence for the cities in YRB. Furthermore, although industrial upgrading has also played a significant role in promoting convergence, its impact is weaker than other conditions.

As far as the status quo of industrial development and emission reduction synergy for cities in the YRB is concerned at this stage, the government should not only pay attention to the “inter-structure upgrading dividend” released by the upgrading of industrial structure from industry to service industry but also pay more attention to the “intra-structure optimization dividend” of industrial internal optimization and adjustment: that is, promoting cross-industry adjustment steadily, focusing on the extension of the industrial chain within the same industry, improvement of factor input efficiency, and optimization of output value. Additionally, the expansion of innovation output and improvement of innovation capability also showed a significant convergence-promoting effect, and the convergence rate increased by 67.38% after taking into account the conditions of industrial optimization and upgrading, indicating that the improvement of urban innovation capability further expanded the role of emission reduction brought by the optimization and upgrading of industrial structure. The government’s attention to green development has also had a significant impact on the carbon emission reduction process. Under the condition that all three mechanisms are included at the same time, the CEI convergence rate for all cities has further increased to 69.96%, indicating that the government’s emphasis on green development and inclination to low-carbon policies are also effective measures towards carbon emission reduction.

As to each urban agglomeration (Table 7), first of all, the conditional β convergence of urban agglomerations in the YRB is significant, and, compared with the rate of absolute β convergence, the rate of conditional β convergence has obviously increased, indicating that the included conditions have a significant impact on the improvement of the convergence rate of CEI at any urban agglomeration in the YRB. Secondly, the spatial correlation model has partially changed. Specifically, the spatial model of SDP has degenerated from SDM to SEM, and that of JZ has changed from SEM to SDM. However, the direction of the spatial effect has not changed, indicating that the positive spatial spillover effect of CEI still exists significantly.

In addition, the rate of conditional β convergence of urban agglomerations in the study period varies. Specifically, among urban agglomerations in the midstream and downstream of the Yellow River, the convergence rates of the three urban agglomerations are similar, including SDP, CP, and GZP. The increase in the rate of absolute β convergence is the largest for GZP, while the absolute value of the convergence rate is the largest for JZ. Hence, the included conditions have a strong convergence-promoting effect on the GZP and JZ, with large regional differences in CEI, while the impact on SDP and CP, with low CEI, is relatively low. Among the upstream urban agglomerations, the convergence rate and its increase are both large for HBEY and LX, while the convergence rate for NX is the slowest among the seven urban agglomerations, indicating that the convergence conditions in NX play a lesser role in promoting convergence than in other urban agglomerations.

In terms of the specific impact of each convergence condition, first, the facilitation of convergence of urban innovation and the government’s attention to green development is more stable, and this is more obvious in the middle and upper reaches of urban agglomerations. On the one hand, urban agglomerations in the middle and upper reaches of the YRB should pay more attention to improving urban innovation capacity and inject new impetus into the low-carbon transformation of urban development by encouraging the expansion of innovation output. On the other hand, the middle and upper reaches are located in the fragile ecological environment zone of the YRB. Promoting the protection of the ecological environment not only conforms to development’s logic and requirements but also significantly promotes the process of carbon emission reduction. Secondly, industrial upgrading and optimization are significantly beneficial in the process of promoting convergence, but the impacts are different in urban agglomerations. The role of industrial upgrading in carbon emission reduction is significantly higher than that of industrial rationalization in the downstream of the YRB, while the role is the opposite in the upstream, and shows a similar coefficient effect in the midstream urban agglomerations.

The above results mainly reveal that development differences in urban agglomerations in the YRB will affect the main direction of industrial structure adjustment. First, SDP and CP should be committed to promoting the upgrading of industrial structure; steadily promoting the transfer of low-value-added industries in the region; actively cultivating high-tech industries and promoting the systematic development of high-quality service industries; and maintaining the stable achievement of carbon control goals. Secondly, on the basis of actively undertaking the transfer of downstream industries, GZP and JZ should concentrate on the internal optimization and adjustment of local industries, accelerating the industrial adjustment process of surrounding cities with measures of industrial optimization of provincial capitals, actively guiding the optimization of regional labor allocation, and pushing forward the continuous promotion of the carbon emission reduction process. Thirdly, NX, HBEY, and LX are the key regions for ecological protection and carbon emission reduction in the YRB. These three urban agglomerations should focus on the problems of low efficiency and high energy consumption within the industry to promote the transformation of industrial structure, green transformation, and intelligent transformation, gradually solving the problems left over by history on the basis of ensuring that the ecological environment does not deteriorate, so as to prevent the carbon emission reduction process from being blocked.

#### 3.3.4. Robustness Test

In order to ensure the robustness of the regression results mentioned above, this section makes a re-examination of conditional β convergence by changing the spatial weight matrix and estimation method. Firstly, on the basis of keeping the process of the test unchanged, the geographical distance matrix is replaced by the geographical proximity matrix (W_0–1_) and the economic geographical distance matrix (W_e_) for regression (Table 8). Under the condition of changing the spatial weight matrix, the characteristics of the conditional β convergence of CEI of urban agglomerations in the YRB still exist significantly, and the direction of the coefficient is consistent with the previous results. Although the spatial effect mode has changed partially, there is still a significant positive spatial spillover effect on the growth rate of CEI. In addition, although the convergence rate of CEI for each urban agglomeration has changed, the coefficient differences between urban agglomerations are basically consistent with the previous results. It can be preliminarily explained that the above regression results have a certain robustness.

Additionally, considering that variable selection and matrix setting may lead to endogeneity problems, the generalized spatial two-stage least-squares (GS2SLS) method combined with the instrumental variable method is used for further regression [52]. On the one hand, in order to avoid endogenous problems caused by economic factors and remain consistent with the benchmark regression, the geographical distance matrix is set as the spatial weight matrix in this part. On the other hand, it is difficult to find instrumental variables satisfying the requirements of correlation, exogeneity, and exclusivity for each explanatory variable. In view of this, the multi-order spatial lag term of the explanatory variable (the lag order is set to 3 in this paper to avoid the problem of weak instrumental variables) is used as the instrumental variable of the spatial lag term of the explained variable for regression. In addition, in order to deal with the endogenous problem caused by reverse causality between the growth rate of CEI and other explanatory variables, as well as considering the lag effects of various convergence conditions, the regression is conducted with the explanatory variable of one-period lag.

From the results of the GS2SLS estimation method (Table 9), the F-statistic value of the regression equation in the first stage of each model is greater than 10, which indicates no obvious problem of weak instrumental variables. From the specific regression results, there are still significant positive spatial spillover effects and conditional β convergence characteristics for the CEI of urban agglomerations in the YRB, and the direction and significance level of the convergence coefficients have not changed significantly. In addition, although the convergence rate of the CEI of each urban agglomeration fluctuates, the difference level among urban agglomerations is basically consistent with the above results, and all these indicate that the regression results of conditional β convergence are robust.

## 4. Conclusions and Suggestions

### 4.1. Conclusions

Based on the CEADs database, this paper calculates the CEI of urban agglomerations in the YRB, and spatial Markov chain, Dagum’s Gini coefficient, and spatial convergence models are used to investigate the distribution dynamics, regional differences, and spatial convergence of CEI for urban agglomerations in the YRB. Additionally, this paper focuses on the differential impacts of urban innovation; industrial optimization and upgrading; and government’s attention to green development on promoting the convergence of CEI in different urban agglomerations.

Firstly, the significant spatial agglomerative and spatial spillover effects of CEI of urban agglomerations in the YRB are confirmed, but these effects did not change significantly during the study period. The time accumulation and path-locking effects of neighborhood-type transitions are also obvious; the possibilities of cross-stage transfers and cross-space transfers of CEI are small. This shows that the types of CEI in cities in the YRB are not isolated in space, and the CEI of a city is significantly related to its neighborhood’s type. Relying on the neighborhood-driven approach can increase the type-transfer probability for high-carbon-emission cities in the YRB; however, the driving effect is limited in general.

Secondly, the CEI of urban agglomerations in the YRB is declining, but overall differences continue to rise. The main source of these overall differences is the differences between urban agglomerations, but the level of differences within urban agglomerations cannot be ignored. The differences within urban agglomerations mainly come from the fact that the CEI of provincial capital cities is significantly lower than that of other cities, which is particularly obvious in GZP and JZ. The differences between urban agglomerations mainly come from connatural differences in the carbon emissions of urban agglomerations. The CEI of the downstream urban agglomerations is much lower than that of the upstream and midstream urban agglomerations, which essentially reflects the developmental differences between the YRB urban agglomerations.

Thirdly, promoting technological innovation; accelerating industrial optimization and upgrading; and enhancing the government’s attention to green development will have a significant impact on the convergence of CEI in urban agglomerations in the YRB. Among these, the effect of enhancing technological innovation and increasing the government’s attention to green development in promoting convergence is relatively stable, and the effect in the middle and upper reaches of urban agglomerations is more obvious. In addition, compared with promoting innovation and strengthening the focus on green development, the effect of industrial optimization and upgrading is more significant, which also shows a significant upstream, middle, and downstream differentiation in promoting the convergence of CEI. Downstream urban agglomerations can promote the decline of CEI by promoting the rationalization of industrial structure, and upstream urban agglomerations should focus on promoting the upgrading of industrial structure, while midstream urban agglomerations need to consider both rationalization and upgrading.

### 4.2. Suggestions

Urban agglomerations in the YRB cover a wide area with complex geographical conditions, and there is a large gap in the development of cities in the upper, middle, and lower reaches. The unbalanced and insufficient development in the region is prominent, leading to different paces of carbon emissions reduction among different urban agglomerations. In the future, the overall direction of promoting carbon emission reduction in urban agglomerations of the YRB should be reducing the differences in urban agglomerations on the basis of maintaining the downward trend of carbon emissions from rebounding. On the one hand, efforts should be made to promote carbon emission reduction within urban agglomerations. On the basis of continuing to play the pioneering role of provincial capitals, efforts should be made to promote the low-carbon transformation process of high-carbon-emission cities, improve the status of regional polarization, and effectively mitigate differences within urban agglomerations. On the other hand, the government should give full play to the synergistic role of Shandong, Henan, and other places with satisfying carbon emission control, and release the potential of green development as well as quality improvement of urban agglomerations.

Furthermore, considering that the development of urban agglomerations in the YRB is faced with such problems as a lower developmental level; a large proportion of resource-based industries and traditional manufacturing industries; and difficulties in the transformation of new and old kinetic energy, each urban agglomeration in the YRB should choose different development strategies in the process of promoting the optimization and upgrading of industrial structure. Among these, downstream urban agglomerations should actively promote the transfer process for traditional industries; vigorously cultivate high-end manufacturing, high-tech industries, and modern service industries with high added value and systematic development potential; and promote the upgrading of regional development and low-carbon transformation by optimizing the industrial structure. Midstream urban agglomerations should be prepared for industrial upgrading and can play a necessary role in guiding industrial optimization at the same time, in order to help the continuous progress of carbon emission reduction. Upstream urban agglomerations should focus on local industries with high energy consumption and low efficiency and deepen and promote the green transformation process with technological progress and rational allocation of factors. In this process, local governments should focus on promoting innovation constantly and strengthen their attention to green and sustainable development. On the one hand, urban agglomerations in the YRB should be committed to taking the path of innovation-driven and intelligence-driven models to improve the overall development level of urban agglomerations. On the other hand, urban agglomerations in the YRB should ensure the quality of green development in urban agglomerations based on the basic requirements of ecological environment protection.

Urban agglomerations in the YRB are areas of concentrated population and core areas of high-quality economic development of China. Promoting the developmental optimization and low-carbon transformation of urban agglomerations in the YRB not only reflects the strategic consideration of the sustainable development of China, but also is an important support for the regional coordinated development strategy to achieve the integration of the whole nation. In the future, based on differences in urban agglomerations in the upper, middle, and lower reaches of the YRB in terms of geographic conditions, natural endowments, and developmental foundations, China should enhance its innovation capability, accelerate the optimization and upgrading of industries, and strengthen the government’s attention to green development. In this way, the spatial coordination and complementary development capacity of urban agglomerations in the YRB can be improved, the process of carbon emission reduction can be steadily promoted, and high-quality sustainable development can be achieved.

## Figures and Tables

**Figure 1 ijerph-20-03529-f001:**
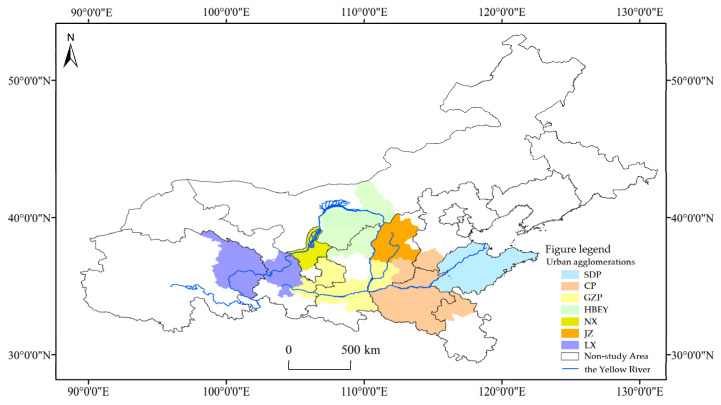
Scope of urban agglomerations in the Yellow River basin.

**Figure 2 ijerph-20-03529-f002:**
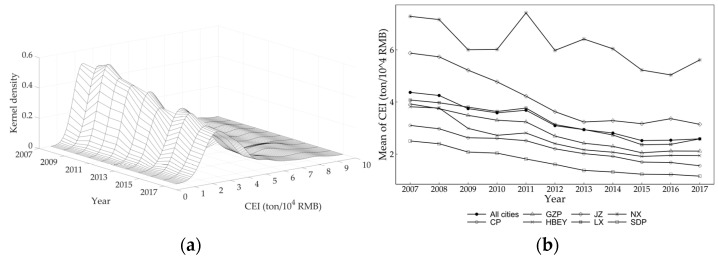
Temporal evolution characteristics of the CEI of urban agglomerations in the Yellow River basin from 2007 to 2017. (**a**) Kernel density curve. (**b**) Changes in average CEI.

**Figure 3 ijerph-20-03529-f003:**
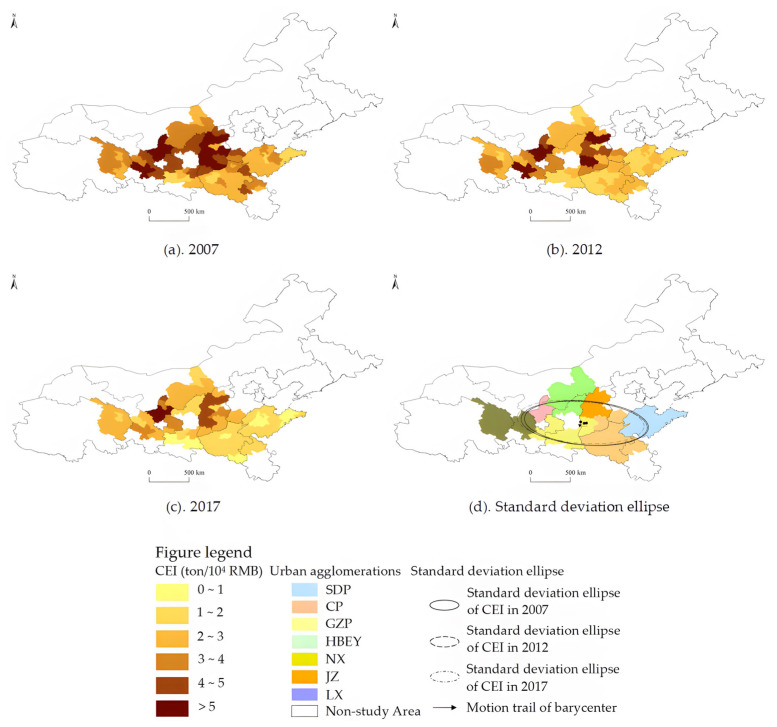
Spatial evolution characteristics of the CEI of cities in the YRB from 2007 to 2017.

**Figure 4 ijerph-20-03529-f004:**
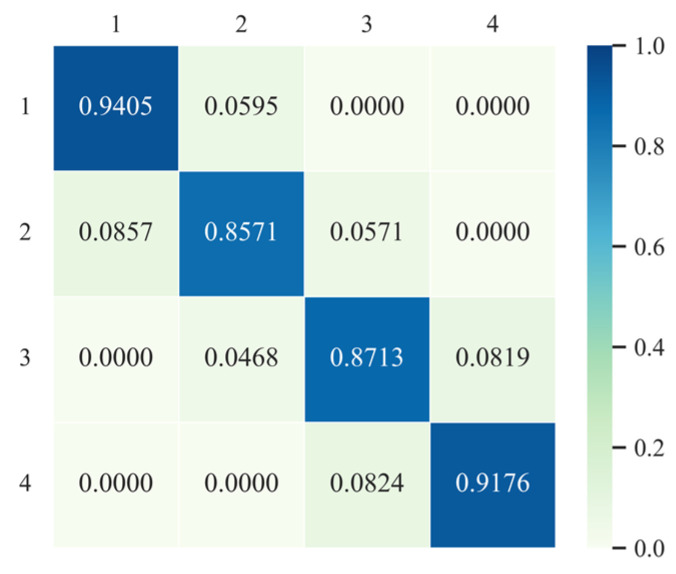
Markov transition probability matrix for CEI types at city level in the YRB from 2007 to 2017.

**Figure 5 ijerph-20-03529-f005:**
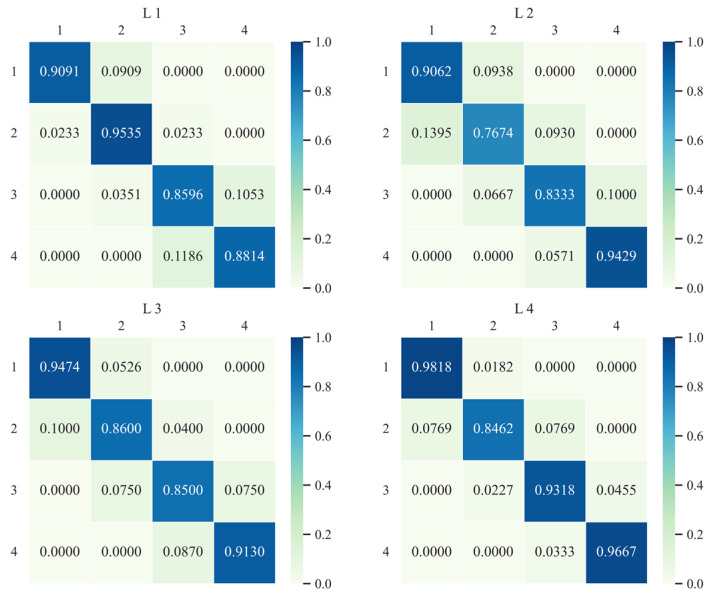
Spatial Markov transition probability matrix for CEI types at city level in the YRB from 2007 to 2017.

**Figure 6 ijerph-20-03529-f006:**
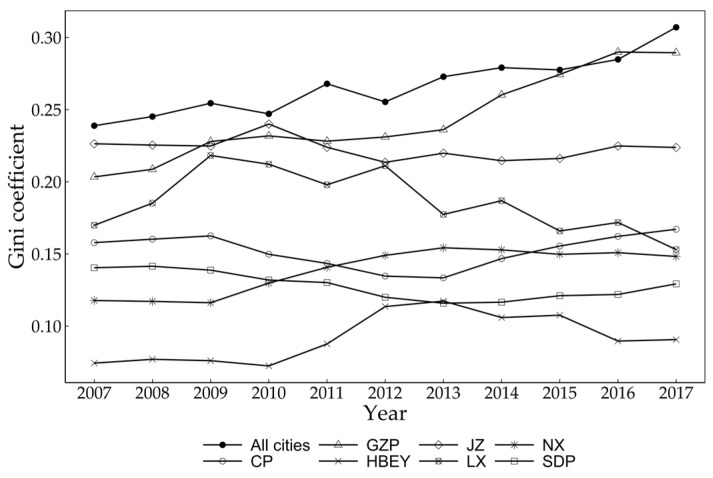
Changes in internal differences in the CEI of urban agglomerations in the YRB from 2007 to 2017.

**Figure 7 ijerph-20-03529-f007:**
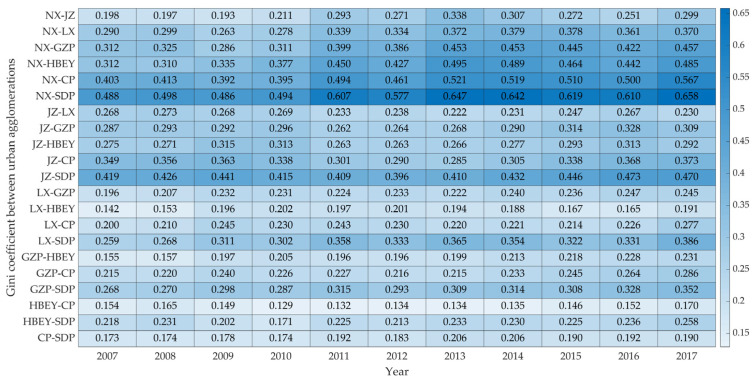
Changes in CEI differences between urban agglomerations in the YRB from 2007 to 2017.

**Figure 8 ijerph-20-03529-f008:**
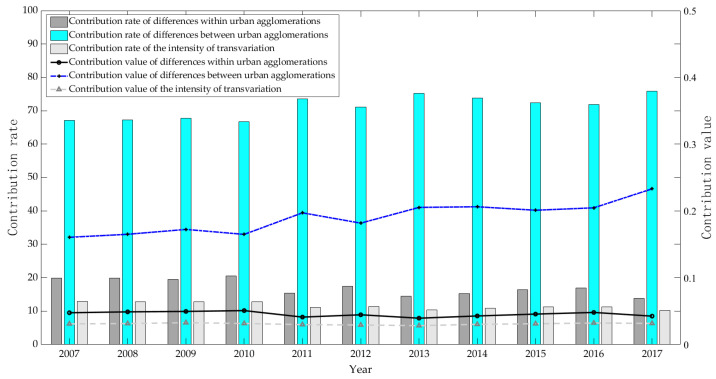
Evolution of contribution to CEI’s differences of urban agglomerations in the YRB from 2007 to 2017.

**Figure 9 ijerph-20-03529-f009:**
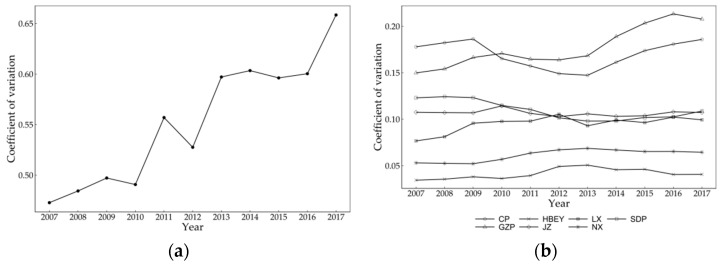
The evolutionary trend of σ convergence of CEI of urban agglomerations in the YRB from 2007 to 2017. (**a**) All cities. (**b**) Urban agglomerations.

**Table 1 ijerph-20-03529-t001:** Descriptive statistics of main variables.

Variables	Interpretation	Mean	Std. Dev.	Min	Max
CEI (CEI)	Total carbon emissions per unit GDP (Ton/10^4^ RMB)	2.74	1.58	0.71	9.97
Urban Innovation (INNOV)	Urban Innovation Index	4.48	11.90	0.01	141.48
Industrial Upgrading (ISU)	The industrial structure upgrading index	2.24	0.13	1.95	2.65
Industrial Optimization (ISR)	The industrial structure rationalization index	0.09	0.30	0.01	4.88
Government’s Attention to Green Development (AGD)	Relative words frequency ratio	1.00	0.34	0.16	2.40
Economic Development (PRGDP)	GDP per unit population (10^4^ RMB)	3.59	2.72	0.34	18.01
Population Density (PD)	Population per unit area (10^3^ person)	0.48	0.28	0.02	1.36
Road Area Ratio (RAR)	Ratio of road area to built-up area	0.17	0.08	0.01	1.13

**Table 2 ijerph-20-03529-t002:** Global Moran’s I of CEI in the YRB from 2007 to 2017.

Year	Moran’s I	Z-Statistic	*p*-Value	Year	Moran’s I	Z-Statistic	*p*-Value
2007	0.14	8.83	0.00	2013	0.16	10.23	0.00
2008	0.14	9.02	0.00	2014	0.15	9.91	0.00
2009	0.14	9.01	0.00	2015	0.14	9.33	0.00
2010	0.13	8.28	0.00	2016	0.14	9.27	0.00
2011	0.16	10.37	0.00	2017	0.16	10.38	0.00
2012	0.15	9.40	0.00				

**Table 3 ijerph-20-03529-t003:** LM test of spatial panel (absolute β convergence).

	Moran’s I		Spatial Error				Spatial Lag			
	Statistic	*p* Value	LM		Robust LM		LM		Robust LM	
			Statistic	*p* Value	Statistic	*p* Value	Statistic	*p* Value	Statistic	*p* Value
All cities	59.18	0.00	203.25	0.00	36.54	0.00	182.22	0.0	15.51	0.00
SDP	25.37	0.00	44.79	0.00	26.93	0.00	22.05	0.00	4.20	0.04
CP	13.97	0.00	70.71	0.00	13.71	0.00	64.82	0.00	7.81	0.01
GZP	32.02	0.00	97.42	0.00	21.01	0.09	77.42	0.00	11.26	0.04
JZ	4.43	0.00	16.48	0.00	2.41	0.02	18.45	0.00	4.37	0.04
NX	6.43	0.00	36.43	0.00	5.12	0.12	40.63	0.00	9.32	0.00
HBEY	5.15	0.00	21.01	0.00	0.13	0.22	23.84	0.00	2.96	0.09
LX	3.94	0.00	12.80	0.00	1.20	0.07	13.11	0.00	1.51	0.02

**Table 4 ijerph-20-03529-t004:** Absolute β convergence characteristics of CEI for urban agglomerations in the YRB.

Region	All Cities	SDP	CP	GZP	JZ	NX	HBEY	LX
Model	SDM	SDM	SDM	SEM	SEM	SAR	SAR	SAR
*β*	−0.208 ***	−0.157 ***	−0.148 ***	−0.083 ***	−0.166 **	−0.214 *	−0.086 ***	−0.084 *
	(−7.879)	(−4.306)	(−4.990)	(−1.111)	(−2.026)	(−1.891)	(−2.615)	(−1.454)
*ρ or λ*	0.922 ***	0.874 ***	0.862 ***	0.772 ***	0.533 ***	0.770 ***	0.593 ***	0.487 ***
	−99.191	−47.752	−34.247	−33.533	−11.213	−18.111	−10.278	−5.392
*γ*	0.201 ***	0.151 ***	0.148 ***	—	—	—	—	—
	−8.273	−4.175	−5.231					
*sigma2_e*	0.002 ***	0.001 ***	0.001 ***	0.002 ***	0.003 **	0.002 ***	0.003 ***	0.005 ***
	−7.586	−5.002	−3.983	−6.209	−2.166	−4.917	−4.529	−2.863
*v*	0.0233	0.0171	0.016	0.0087	0.0182	0.0241	0.009	0.0088
Hausman Test (*p* value)	94.79 (0.000)	19.67 (0.000)	21.15 (0.000)	5.35 (0.021)	4.34 (0.034)	7.14 (0.008)	13.80 (0.000)	2.69 (0.011)
Wald Test (*p* Value)	68.45 (0.000)	17.43 (0.000)	27.37 (0.000)	1.89 (0.169)	25.08 (0.000)	3.29 (0.070)	2.70 (0.101)	0.49 (0.084)
LR Test_SAR (*p* Value)	63.75 (0.000)	17.61 (0.000)	17.88 (0.000)	12.20 (0.034)	2.76 (0.097)	2.43 (0.327)	7.00 (0.218)	6.44 (0.231)
LR Test_SEM (*p* Value)	39.86 (0.027)	2.82 (0.093)	3.71 (0.054)	3.24 (0.621)	2.13 (2.289)	1.63 (0.019)	1.52 (0.001)	1.55 (0.059)
Time Effect	YES	YES	YES	YES	YES	YES	YES	YES
Individual Effect	YES	YES	YES	YES	YES	YES	YES	YES

Note: ***, **, and * indicate significant correlations at the 0.01, 0.05, and 0.10 levels, respectively.

**Table 5 ijerph-20-03529-t005:** LM test of spatial panel (conditional β convergence).

	Moran’s I		Spatial Error				Spatial Lag			
	Statistic	*p* Value	LM		Robust LM		LM		Robust LM	
			Statistic	*p* Value	Statistic	*p* Value	Statistic	*p* Value	Statistic	*p* Value
All cities	58.22	0.00	3009.15	0.000	54.10	0.000	3059.66	0.00	51.05	0.00
SDP	24.51	0.00	525.70	0.000	25.06	0.000	507.615	0.00	6.98	0.09
CP	32.35	0.00	882.00	0.00	12.54	0.00	879.60	0.00	10.14	0.00
GZP	12.81	0.00	140.27	0.00	1.51	0.02	153.43	0.00	14.67	0.04
JZ	3.310	0.00	8.41	0.00	9.87	0.00	15.17	0.00	16.62	0.00
NX	5.25	0.00	22.35	0.00	0.00	0.98	31.99	0.00	9.64	0.00
HBEY	3.98	0.00	14.22	0.00	0.26	0.61	20.94	0.00	6.98	0.01
LX	2.25	0.02	4.56	0.03	5.19	0.08	9.13	0.00	9.76	0.00

**Table 6 ijerph-20-03529-t006:** Conditional β convergence characteristics of CEI of all cities in the YRB.

Model	SDM (1)	SDM (2)	SDM (3)	SDM (4)	SDM (5)	SDM (6)	SDM (7)
*β*	−0.285 ***	−0.267 ***	−0.313 ***	−0.321 ***	−0.323 ***	−0.272 ***	−0.327 ***
	(−8.016)	(−8.000)	(−6.058)	(−6.247)	(−6.177)	(−7.957)	(−6.177)
*ρ*	0.885 ***	0.879 ***	0.875 ***	0.865 ***	0.863 ***	0.890 ***	0.863 ***
	(40.392)	(36.544)	(39.036)	(34.323)	(33.432)	(42.929)	(33.432)
*γ*	0.115 **	0.118 *	0.134 *	0.008 *	0.005 *	0.162 **	0.006 *
	(1.464)	(1.892)	(1.794)	(0.081)	(0.060)	(2.157)	(0.065)
*INNOV*	−0.001 ***				−0.001 ***		−0.001 ***
	(−3.264)				(−3.356)		(−3.356)
*ISU*		−0.006 *		−0.010	−0.009		−0.009
		(−2.060)		(−0.093)	(−0.075)		(−0.075)
*ISR*			−0.007 **	−0.008 **	−0.006 **		−0.006 **
			(−1.113)	(−1.096)	(−0.900)		(−0.900)
*AGD*						−0.003 **	−0.003 **
						(−0.560)	(−0.481)
*sigma2_e*	0.002 ***	0.002 ***	0.002 ***	0.002 ***	0.002 ***	0.002 ***	0.002 ***
	(8.124)	(8.194)	(8.779)	(8.728)	(8.652)	(8.220)	(8.652)
*v*	0.0335	0.0311	0.0375	0.0387	0.0390	0.0317	0.0396
Hausman Test (*p* Value)	133.96 (0.000)	124.31 (0.000)	144.96 (0.000)	139.39 (0.000)	145.19 (0.000)	125.83 (0.000)	144.53 (0.000)
Wald Test (*p* Value)	49.24 (0.000)	31.38 (0.000)	44.97 (0.000)	40.82 (0.000)	45.14 (0.000)	37.59 (0.000)	54.51 (0.000)
LR Test_SAR (*p* Value)	53.09 (0.000)	18.45 (0.000)	64.88 (0.000)	60.36 (0.000)	62.97 (0.000)	49.54 (0.000)	69.37 (0.000)
LR Test_SEM (*p* Value)	36.09 (0.000)	12.61 (0.006)	47.76 (0.000)	50.07 (0.000)	53.55 (0.000)	34.34 (0.000)	14.84 (0.011)
Control Variable	YES	YES	YES	YES	YES	YES	YES
Time Effect	YES	YES	YES	YES	YES	YES	YES
Individual Effect	YES	YES	YES	YES	YES	YES	YES

Note: ***, **, and * indicate significant correlations at the 0.01, 0.05, and 0.10 levels, respectively.

**Table 7 ijerph-20-03529-t007:** Conditional β convergence characteristics of CEI of urban agglomerations in the YRB.

Region	SDP	CP	GZP	JZ	NX	HBEY	LX
Model	SEM	SDM	SEM	SDM	SAR	SAR	SAR
*β*	−0.371 ***	−0.382 ***	−0.398 ***	−0.567 ***	−0.331 ***	−0.550 ***	−0.585 ***
	(−3.264)	(−5.001)	(−3.754)	(−4.386)	(−4.201)	(−2.646)	(−4.709)
*ρ or λ*	0.869 ***	0.773 ***	0.686 ***	0.405 ***	0.716 ***	0.535 ***	0.407 ***
	(42.487)	(14.885)	(18.840)	(5.106)	(13.783)	(7.209)	(4.077)
*γ*	-	0.049 *	-	0.861 ***	-	-	-
		(0.342)		(2.727)			
*INNOV*	−0.001 **	−0.003 ***	−0.001	−0.022 ***	−0.003	−0.023 *	−0.016 *
	(−2.140)	(−3.269)	(−1.363)	(−6.115)	(−0.284)	(−1.677)	(−1.800)
*ISU*	−0.564 **	−0.332 ***	−0.456 **	−0.372 ***	−0.003	−0.009	−0.035
	(−2.308)	(−3.006)	(−2.283)	(−3.099)	(−0.035)	(−0.038)	(−0.864)
*ISR*	−0.017 **	−0.006 *	−0.487 *	−0.052 **	−0.685	−0.073 **	−0.303 *
	(−0.248)	(−0.026)	(−1.706)	(−0.184)	(−0.639)	(−2.414)	(−0.464)
*AGD*	−0.003 *	−0.009 *	−0.016 **	−0.031 *	−0.041 *	−0.040	−0.036 *
	(−0.379)	(−1.472)	(0.995)	(−1.261)	(−1.892)	(0.730)	(−0.980)
*sigma2_e*	0.000 ***	0.001 ***	0.002 ***	0.001 **	0.002 ***	0.002 ***	0.002 **
	(6.222)	(4.401)	(7.437)	(2.378)	(5.141)	(5.307)	(2.257)
*v*	0.0464	0.0481	0.0507	0.0837	0.0402	0.0799	0.0879
Hausman Test (*p* Value)	40.16 (0.000)	41.33 (0.000)	32.16 (0.000)	21.80 (0.005)	9.79 (0.073)	11.53 (0.073)	22.81 (0.004)
Wald Test (*p* Value)	18.16 (0.039)	21.76 (0.005)	20.35 (0.009)	12.51 (0.013)	10.92 (0.012)	10.09 (0.018)	7.92 (0.048)
LR Tesy_SAR (*p* Value)	18.16 (0.020)	23.76 (0.003)	9.56 (0.078)	13.70 (0.090)	17.37 (0.265)	10.52 (0.230)	8.07 (0.426)
LR Test_SEM (*p* Value)	7.70 (0.463)	13.90 (0.084)	6.62 (0.297)	14.08 (0.080)	15.63 (0.048)	14.44 (0.071)	9.67 (0.049)
Control Variable	YES	YES	YES	YES	YES	YES	YES
Time Effect	YES	YES	YES	YES	YES	YES	YES
Individual Effect	YES	YES	YES	YES	YES	YES	YES

Note: ***, **, and * indicate significant correlations at the 0.01, 0.05, and 0.10 levels, respectively.

**Table 8 ijerph-20-03529-t008:** Robustness tests based on different spatial weight matrices.

Region	All Cities		SDP		CP		GZP	
Matrix	W_0–1_	W_e_	W_0–1_	W_e_	W_0–1_	W_e_	W_0–1_	W_e_
Model	SDM	SDM	SDM	SEM	SDM	SAR	SEM	SEM
*β*	−0.315 ***	−0.338 ***	−0.304 **	−0.369 ***	−0.398 ***	−0.348 ***	−0.420 ***	−0.511 ***
	(−4.173)	(−5.161)	(−2.516)	(−3.607)	(−4.656)	(−5.736)	(−3.780)	(−6.594)
*ρ or λ*	0.802 ***	0.379 ***	0.512 ***	0.551 ***	0.729 ***	0.482 ***	0.649 ***	0.364 ***
	(24.610)	(9.884)	(8.091)	(8.382)	(12.467)	(10.486)	(12.299)	(4.471)
*γ*	−0.064	−0.084	−0.722 ***	—	0.158	—	—	—
	(−0.698)	(−1.262)	(−3.718)		(1.042)			
*INNOV*	−0.001 ***	−0.001 ***	−0.001 ***	−0.002 ***	−0.002 ***	−0.003 ***	−0.001 **	−0.001 *
	(−3.028)	(−2.764)	(3.299)	(0.068)	(−3.185)	(−2.852)	(−1.395)	(−1.857)
*ISU*	−0.027 **	−0.058 **	−0.311 ***	−0.334 **	−0.375 ***	−0.113 **	−0.414 **	−0.535 **
	(−0.254)	(0.526)	(−3.733)	(−2.036)	(−2.927)	(−1.522)	(−2.059)	(−2.430)
*ISR*	−0.013 **	−0.009 *	−0.110 **	−0.180 *	−0.086 ***	−0.014 **	−0.520 *	−0.422 *
	(1.591)	(−0.973)	(−0.294)	(1.943)	(−0.354)	(−0.380)	(−1.833)	(−1.355)
*AGD*	−0.011 **	−0.004 *	0.127 ***	−0.001 *	−0.027 *	−0.009 *	−0.011 **	−0.028 *
	(−1.297)	(−0.592)	(2.907)	(−0.110)	(−1.834)	(−1.393)	(−1.727)	(−0.917)
*sigma2_e*	0.002 ***	0.003 ***	0.000 ***	0.001 ***	0.001 ***	0.002 ***	0.002 ***	0.002 ***
	(8.401)	(7.700)	(5.713)	(4.990)	(4.252)	(5.746)	(6.964)	(7.371)
*v*	0.0378	0.0412	0.0362	0.0460	0.0507	0.0423	0.0545	0.0715
Hausman Test (*p* Value)	124.232 (0.000)	194.483 (0.000)	162.273 (0.000)	62.298 (0.000)	88.181 (0.000)	75.023 (0.000)	51.246 (0.000)	21.529 (0.006)
Wald Test (*p* Value)	31.421 (0.000)	32.991 (0.000)	58.644 (0.000)	19.271 (0.014)	27.236 (0.001)	22.571 (0.001)	76.543 (0.000)	62.944 (0.000)
LR Test_SAR (*p* Value)	33.447 (0.000)	37.931 (0.000)	53.160 (0.000)	10.290 (0.045)	22.976 (0.000)	9.482 (0.304)	57.849 (0.000)	18.364 (0.040)
LR Test_SEM (*p* Value)	26.755 (0.000)	35.823 (0.000)	43.692 (0.000)	5.871 (0.662)	14.572 (0.068)	16.458 (0.036)	61.443 (0.103)	9.001 (0.343)
Control Variable	YES	YES	YES	YES	YES	YES	YES	YES
Time Effect	YES	YES	YES	YES	YES	YES	YES	YES
Individual Effect	YES	YES	YES	YES	YES	YES	YES	YES
**Region**	**JZ**		**NX**		**HBEY**		**LX**	
**Matrix**	**W_0−1_**	**W_e_**	**W_0−1_**	**W_e_**	**W_0−1_**	**W_e_**	**W_0−1_**	**W_e_**
**Model**	**SDM**	**SEM**	**SAR**	**SAR**	**SAR**	**SAR**	**SAR**	**SAR**
*β*	−0.528 ***	−0.534 ***	−0.272 ***	−0.283 ***	−0.521 ***	−0.567 ***	−0.581 ***	−0.593 ***
	(−4.786)	(−5.689)	(−8.577)	(−7.895)	(−2.640)	(−4.644)	(−5.629)	(−5.597)
*ρ or λ*	0.482 ***	0.497 ***	0.734 ***	0.317 ***	0.557 ***	0.407 ***	0.378 ***	0.306 ***
	(6.480)	(6.877)	(13.659)	(2.791)	(6.504)	(4.449)	(3.905)	(3.255)
*γ*	0.697 ***	—	—	—	—	—	—	—
	(6.508)							
*INNOV*	−0.033 ***	−0.021 ***	−0.001 **	−0.041 *	−0.023 **	−0.039 *	−0.014 **	−0.016 **
	(−9.791)	(−4.933)	(−0.135)	(−1.782)	(−1.632)	(−1.679)	(−2.126)	(−2.151)
*ISU*	−0.305 **	−0.522 **	−0.102 *	−0.014 **	−0.021 *	−0.148 **	−0.124 *	−0.035 *
	(−2.506)	(−1.049)	(−0.278)	(−0.915)	(−1.151)	(−2.446)	(0.429)	(−0.191)
*ISR*	−0.335 ***	−0.470 *	−0.622 **	−0.488 *	−0.069 **	−0.353 **	−0.588 **	−0.303 **
	(−3.257)	(−0.952)	(−0.646)	(−1.690)	(−2.351)	(−0.896)	(−1.209)	(−0.554)
*AGD*	−0.110 ***	0.026 **	−0.039 *	−0.099 ***	−0.058 **	−0.042 **	−0.049 *	−0.036 *
	(−3.436)	(−0.953)	(−1.829)	(−4.800)	(−0.602)	(−0.761)	(−1.800)	(−1.171)
*sigma2_e*	0.001 ***	0.002 ***	0.001 ***	0.006 ***	0.002 ***	0.004 ***	0.002 ***	0.003 ***
	(10.892)	(2.660)	(6.321)	(2.733)	(5.180)	(5.054)	(4.396)	(4.472)
*v*	0.0751	0.0764	0.0317	0.0333	0.0736	0.0837	0.0870	0.0899
Hausman Test (*p* Value)	18.050 (0.042)	6.530 (0.015)	10.920 (0.062)	12.627 (0.012)	10.517 (0.037)	15.493 (0.050)	16.314 (0.038)	32.856 (0.000)
Wald Test (*p* Value)	31.311 (0.000)	39.826 (0.000)	39.386 (0.000)	47.188 (0.000)	52.336 (0.000)	56.952 (0.000)	37.391 (0.000)	32.025 (0.000)
LR Test_SAR (*p* Value)	41.801 (0.000)	10.953 (0.024)	23.824 (0.302)	6.458 (0.368)	26.921 (0.170)	28.813 (0.219)	26.940 (0.101)	20.716 (0.265)
LR Test_SEM (*p* Value)	42.813 (0.000)	8.920 (0.206)	32.063 (0.003)	8.701 (0.013)	30.460 (0.002)	36.001 (0.001)	29.487 (0.000)	23.197 (0.008)
Control Variable	YES	YES	YES	YES	YES	YES	YES	YES
Time Effect	YES	YES	YES	YES	YES	YES	YES	YES
Individual Effect	YES	YES	YES	YES	YES	YES	YES	YES

Note: ***, **, and * indicate significant correlations at the 0.01, 0.05, and 0.10 levels, respectively.

**Table 9 ijerph-20-03529-t009:** Robustness test based on GS2SLS.

Region	All Cities		SDP		CP		GZP	
Model	Benchmark	One-Period Lag	Benchmark	One-Period Lag	Benchmark	One-Period Lag	Benchmark	One-Period Lag
β	−0.231 ***	−0.276 ***	−0.291 ***	−0.260 ***	−0.289 ***	−0.248 ***	−0.311 **	−0.329 ***
	(−8.098)	(−8.651)	(−3.083)	(−3.018)	(−4.429)	(−3.256)	(−2.441)	(−3.052)
*ρ*	0.047 ***	0.046 ***	0.110 ***	0.107 ***	0.063 ***	0.066 ***	0.149 ***	0.158 ***
	(16.072)	(16.618)	(16.391)	(15.595)	(14.930)	(16.143)	(6.935)	(8.106)
*INNOV*	−0.001 *	−0.001 **	−0.000 **	−0.000 **	−0.003 ***	−0.003 **	−0.001 *	−0.001 *
	(−1.765)	(−2.088)	(−1.348)	(−0.923)	(−3.293)	(−2.500)	(−0.946)	(−0.618)
*ISU*	−0.066 ***	−0.015 ***	−0.355 ***	−0.360 **	−0.058 ***	−0.080 **	−0.154 **	−0.197 **
	(−2.759)	(−3.002)	(−2.629)	(−2.252)	(−0.898)	(−1.062)	(−0.963)	(−0.996)
*ISR*	−0.006 ***	−0.009 ***	−0.028 **	−0.033 **	−0.313 **	−0.044 **	−0.559 **	−0.447 **
	(−0.599)	(−0.744)	(−0.288)	(−0.301)	(−1.331)	(−0.168)	(−2.620)	(−2.030)
*AGD*	−0.006 **	−0.007 **	−0.002	−0.001	−0.006 *	−0.007	−0.015	−0.045 **
	(−0.811)	(−0.878)	(−0.088)	(−0.009)	(−0.853)	(−0.853)	(−0.820)	(−2.365)
*v*	0.0263	0.0323	0.0343	0.0301	0.0341	0.0285	0.0373	0.0399
Hausman Test (*p* Value)	89.949 (0.000)	107.607 (0.000)	36.077 (0.041)	28.278 (0.0211)	33.845 (0.000)	21.005 (0.013)	20.927 (0.013)	19.070 (0.025)
Wald Test (*p* Value)	568.183 (0.000)	621.106 (0.000)	585.568 (0.000)	544.047 (0.000)	381.400 (0.000)	407.041 (0.000)	175.243 (0.000)	199.924 (0.000)
Phase1 F (*p* Value)	63.131 (0.000)	69.012 (0.000)	65.063 (0.000)	60.450 (0.000)	42.3776 (0.000)	45.2268 (0.000)	19.471 (0.000)	22.214 (0.000)
Phase2 F (*p* Value)	258.309 (0.000)	276.167 (0.000)	268.655 (0.000)	243.193 (0.000)	222.904 (0.000)	260.599 (0.000)	48.095 (0.000)	65.706 (0.000)
Control Variable	YES	YES	YES	YES	YES	YES	YES	YES
Time Effect	YES	YES	YES	YES	YES	YES	YES	YES
Individual Effect	YES	YES	YES	YES	YES	YES	YES	YES
**Region**	**JZ**		**NX**		**HBEY**		**LX**	
**Model**	**Benchmark**	**One-Period Lag**	**Benchmark**	**One-Period Lag**	**Benchmark**	**One-Period Lag**	**Benchmark**	**One-Period Lag**
β	−0.420 ***	−0.401 ***	−0.232 **	−0.216 **	−0.436 **	−0.560 ***	−0.514 ***	−0.579 ***
	(−5.090)	(−4.329)	(−2.707)	(−2.245)	(−2.280)	(−4.980)	(−3.275)	(−3.736)
*ρ*	0.232 ***	0.246 ***	0.566 ***	0.501 ***	0.267 ***	0.255 ***	0.378 ***	0.463 ***
	(3.501)	(3.243)	(4.971)	(4.496)	(6.443)	(8.573)	(3.645)	(3.519)
*INNOV*	−0.020 **	−0.010 **	−0.005 *	−0.009 *	−0.018 *	−0.016 *	−0.011 *	−0.007
	(−2.123)	(−1.045)	(−0.180)	(−0.281)	(−0.813)	(−1.281)	(−1.372)	(−0.856)
*ISU*	−0.522 *	−0.153 *	−0.041	−0.209 *	−0.105	−0.427	−0.187 *	−0.207
	(−2.000)	(−0.601)	(−0.795)	(−2.663)	(−0.252)	(−1.358)	(−0.527)	(−0.470)
*ISR*	−0.370 *	−0.188 *	−0.002	−0.027 *	−0.045	−0.177 **	−0.700	−0.065
	(−1.840)	(−0.853)	(−0.098)	(−0.526)	(−0.601)	(−2.695)	(−1.265)	(−0.156)
*AGD*	−0.030 *	−0.057	−0.028 **	−0.100 *	−0.036 *	−0.007	−0.045 **	−0.038 *
	(−0.770)	(−1.074)	(−0.432)	(−1.738)	(−0.664)	(−0.125)	(−1.488)	(−0.916)
*v*	0.0545	0.0512	0.0264	0.0243	0.0573	0.0821	0.0722	0.0865
Hausman Test (*p* Value)	26.704 (0.001)	24.215 (0.008)	38.579 (0.022)	25.482 (0.003)	16.842 (0.018)	13.486 (0.038)	22.941 (0.006)	26.384 (0.002)
Wald Test (*p* Value)	193.751 (0.000)	182.544 (0.000)	155.998 (0.000)	179.736 (0.000)	85.975 (0.000)	79.917 (0.000)	66.892 (0.000)	73.433 (0.000)
Phase1 F (*p* Value)	18.860 (0.000)	13.172 (0.000)	17.333 (0.000)	32.639 (0.000)	19.553 (0.000)	12.013 (0.000)	17.433 (0.000)	12.604 (0.000)
Phase2 F (*p* Value)	49.981 (0.000)	39.456 (0.000)	41.509 (0.000)	73.500 (0.000)	54.712 (0.000)	38.212 (0.000)	43.288 (0.001)	30.515 (0.002)
Control Variable	YES	YES	YES	YES	YES	YES	YES	YES
Time Effect	YES	YES	YES	YES	YES	YES	YES	YES
Individual Effect	YES	YES	YES	YES	YES	YES	YES	YES

Note: ***, **, and * indicate significant correlations at the 0.01, 0.05, and 0.10 levels, respectively.

## Data Availability

The data that support the findings of this study are available upon request from the corresponding author.

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
