# Peer review of "Spatiotemporal Dynamic Distribution, Regional Differences and Spatial Convergence Mechanisms of Carbon Emission Intensity: Evidence from the Urban Agglomerations in the Yellow River Basin"

_ijerph, 2023, doi:10.3390/ijerph20043529_

Round 1
Reviewer 1 Report
The manuscript analyzes the spatiotemporal dynamic distribution, regional differences and spatial convergence mechanisms of carbon emission intensity of urban agglomerations in the Yellow River Basin in detail, with complete contents and clear logic. But there are a few things that can be improved.
1.The data range is 2007-2017, why is the data from 2018-2021 not included?
2.Explain in detail the causes of the spatiotemporal distribution characteristics and spatiotemporal transfer characteristics in the urban agglomeration of the Yellow River Basin.
3.Write conclusions and suggestions separately.
4.What is the significance of this study compared with other studies?
Author Response
We sincerely thank you for your insightful comments on this article.
Please see the attachment.

Reviewer 2 Report
Based on the CEADs database, this paper calculates the CEI of urban agglomerations in the YRB, and spatial Markov chain, Dagum’s Gini coefficient, and spatial convergence model are used to investigate the distribution dynamics, regional differences, and spatial convergence of CEI of urban agglomerations in the YRB.
Generally, the analysis and description of data in this article are standardized and meticulous, and if they can be reasonably organized, the results can have a good role in promoting the green and low-carbon development of cities in the YRB region. There are still issues need to be addressed:
1. The methods used by the authors are relatively mature, there are many specific uses in the literature, such as described in citation [34]. Therefore, it is suggested that the article carries out a certain comparative analysis of why these methods are selected and the advantages and disadvantages of other types of methods, which is conducive to clarifying the rationality and scientific nature of the methods.
2. Three carbon reduction mechanisms as conditions to analyze their impact on the spatial convergence and convergence speed of the CEI of the urban agglomerations in the YRB may not enough. Song et al. (2022) studied the YRB regional differences and convergence by using Theil index,σ convergence and absolute β convergence, and the influencing factors were analyzed by using spatial Dubin model, the results show that, the improvement of economic development level, technological innovation level and foreign trade level have a positive impact on the carbon emission efficiency of the Yellow River Basin, while the energy consumption structure and urbanization level have a negative impact. The result of this article means that not only Urban Innovation (a quite new index for analysis), Optimization and Upgrading of Industrial Structure, Government’s Attention to Green Development are the main carbon reduction mechanisms, other factors such as foreign trade level may have positive impact, which will influence the result.
3. Meanwhile, it is recommended to add an analysis with the results of the currently available study it is recommended that the method and results of this article to be discussed and compared with the results of the currently available study such as Song, et al. (2022), to expand the dimensions of the conclusion of this article.
Ref.: SONG Min; ZOU Sujuan. Regional Differences,Convergence and Influencing Factors of Carbon Emissions Efficiency in the Yellow River Basin. Yellow River, 2022, 44(8):8.
Author Response

(The authors gave the same response as above.)
